# An Emphasis on the Role of Long Non-Coding RNAs in Viral Gene Expression, Pathogenesis, and Innate Immunity in Viral Chicken Diseases

**DOI:** 10.3390/ncrna11030042

**Published:** 2025-05-26

**Authors:** Anindita Sarma, Parul Suri, Megan Justice, Raja Angamuthu, Samuel Pushparaj

**Affiliations:** 1Department of Animal Biotechnology, Madras Veterinary College, Tamil Nadu Veterinary and Animal Sciences University, Chennai 600007, India; 2College of Pharmacy and Health Sciences, St. John’s University, Queens, NY 11439, USA; 3Department of Physiological Sciences, Oklahoma State University, Stillwater, OK 74074, USA

**Keywords:** lncRNAs, viral diseases, poultry

## Abstract

The poultry farming industry encounters considerable obstacles stemming from viral diseases, resulting in elevated mortality rates and substantial economic losses. Current research highlights the significant involvement of long non-coding RNAs (lncRNAs) in the interactions between hosts and pathogens by enhancing antiviral responses at different levels, such as the activation of pathogen recognition receptors, as well as through epigenetic, transcriptional, and post-transcriptional modifications. Specific long non-coding RNAs (lncRNAs), including ERL lncRNA, linc-GALMD3, and loc107051710, have been recognized as significant contributors to the antiviral immune response to multiple avian viral pathogens. Understanding the mechanisms by which long non-coding RNAs (lncRNAs) act offers valuable insights into prospective diagnostic and therapeutic approaches aimed at improving disease resistance in poultry. Differentially expressed lncRNAs may also be utilized as biomarkers for both prognosis and diagnosis of avian viral diseases. This review delves into the various roles of long non-coding RNAs (lncRNAs) in the context of viral diseases in chickens, such as avian leukosis, Marek’s disease, infectious bursal disease, avian influenza, infectious bronchitis, and Newcastle disease. It highlights the pivotal role of lncRNAs in the complex dynamics between the host and viral pathogens, particularly their interactions with specific viral proteins. Understanding these interactions may provide valuable insights into the spatial and temporal regulation of lncRNAs, aid in the identification of potential drug targets, and reveal the expression patterns of lncRNA and coding gene transcripts in response to different viral infections in avian species.

## 1. Introduction

Poultry meat has become the most consumed livestock commodity in the world, especially in developing countries. Due to the increase in demand for poultry meat, there has been an increase in global imports during the period from 2001 to 2021. Poultry meat import is going to be the most in the next ten years, and most nations have raised their domestic poultry production to match this need. Major exporters of poultry meat are Brazil, the United States, the European Union, and Thailand. The world’s leading poultry exporter, Brazil, is going to remain in the top position through 2031, as mentioned in The National Registry of Genetically Unique Animal Populations: USDA-ADOL Chicken Genetic Lines USDA, 2022 [1].

The main obstacle to modern poultry farming is viral diseases in the birds. Poultry birds are frequently the targets of viral diseases (some of which include zoonotic diseases), which result in high mortality and morbidity. These diseases are caused by a variety of factors, including the environment, stress, irregular vaccination, nutritional and immune-suppressive additives such as antibiotics, leading antimicrobial resistance in poultry, and a lack of proper biosecurity. This, in turn, has an impact on the economics of poultry production, increasing losses and decreasing gains. Viruses are microorganisms that are inert particles and can only survive in living organisms. In order to avoid being recognized and eliminated by the host immune system, they must take advantage of the host immune system. Viruses create a variety of proteins that resemble crucial host-specific components in order to evade the host immune system, as mentioned in the work of Mohandas et al., 2018 [2]. Reticulo-endotheliosis virus, lymphoid leukosis (LL), infectious bursal disease virus (IBDV), avian influenza virus (AIV), Newcastle disease virus (NDV), Marek’s disease virus (MDV), and chicken infectious anemia virus (CIAV) are some of the avian viruses that significantly cause immunosuppression, as mentioned by Balamurugan and Kataria in 2006 [3]. Viral infections of this nature induce significant lymphoproliferation and infiltration of mononuclear cells throughout various regions of the body. This process can compromise immune defense mechanisms, which in turn adversely impacts growth and overall performance, resulting in pronounced atrophy of lymphoid organs, particularly the bursa and thymus. Additionally, this process has the potential to induce profound immunosuppression, which can compromise the antibody response and elevate the risk of secondary diseases, such as bacterial, protozoal, or parasitic infections. Co-infection of one or more viruses can worsen immunosuppression and accelerate the progression of disease in birds, resulting in symptoms such as decreased feed conversion ratio (FCR) and weight gain, incapability to eat, lethargy, and eventually death. However, in response to a virus’s entry, the host immune system will use a variety of immunoregulation mechanisms to remove the virus from the host. One notable example is long non-coding RNAs in chickens. These lncRNAs are essential for regulating the host immune response against pathogenic infections, thereby facilitating innate immunity. They regulate various immune-related genes, including lincRNA-Cox2 and NEAT1, through interactions with host proteins that are integral to immune responses. Long non-coding RNAs (lncRNAs) play a significant role in the regulation of gene expression through diverse mechanisms. These mechanisms involve immunomodulation, transcriptional and post-transcriptional regulation, modulation of protein activity, and functions related to nuclear organization and scaffolding [4]. A significant number of research studies have explored the immune responses associated with viral infections, particularly examining the mechanisms by which long non-coding RNAs (lncRNAs) contribute to viral pathogenesis. This study highlights the ways in which lncRNAs support the host immune system in evading infections and enhancing innate immunity, and also describes the action of lncRNAs in viral gene expression, as mentioned in detail in this review.

Over the past several years, the chicken genome has been the subject of thorough investigation, especially concerning long non-coding RNAs (lncRNAs) involved not only in immune responses but also in the growth and fertility of poultry. As meat performance represents a significant economic phenotype in poultry, a considerable body of research has been dedicated to exploring lncRNA expression in tissues associated with muscle development [5]. Research conducted by Peng et al. in 2018 examined the role of long non-coding RNAs (lncRNAs) in egg-laying performance [6]. Their study demonstrated that an integrated network analysis, which identified differentially expressed genes, was associated with ovarian follicular development. This development is linked to processes such as oocyte meiosis, progesterone-mediated maturation of oocytes, and the cell cycle, highlighting the potential importance of specific lncRNAs in ovarian follicular development.

Extracellular vesicles (EVs) are membrane-bound structures composed of lipids that are released by cells into the extracellular environment. They serve as carriers for transporting a variety of molecules. The size of extracellular vesicles (EVs) enables them to accommodate a diverse range of molecules and molecular complexes. The process of cargo packaging is dependent on the EV class, with exosomes demonstrating a preferential selection process that incorporates various mechanisms involving highly conserved proteins, including Rab GTPases and endosomal sorting complexes required for transport (ESCRTs). Recent studies have demonstrated that certain viruses can utilize extracellular vesicles (EVs) to facilitate their entry and exit from cells, indicating that EVs play a significant role as carriers of viral material [7]. Limited research has been conducted on the functions of long non-coding RNAs (lncRNAs) within this biological network. Similar to lipids, proteins, DNA, and mRNA, lncRNAs are conveyed in vertebrates through exosomes. However, their roles in vectors or arthropods remain inadequately explored, leaving many of their functions unidentified. It is possible that lncRNAs are expressed in response to pathogen infection of the host or by the vector to mitigate host defense mechanisms. There is a hypothesis that lncRNAs from vectors may be delivered via salivary exosomes to the host, where they could serve as “sponges” for host microRNAs, thereby disrupting natural defense processes [8].

Research focused on the transport of lncRNAs through extracellular vesicles has been conducted to elucidate the functions of lncRNAs in the process of neoplastic transformation, with the overarching aim of creating innovative non-invasive diagnostic methods. While our understanding of the significance of EV-trafficked lncRNAs in the context of viral infections is still new, emerging research indicates their potential role. For example, exosomes extracted from individuals infected with HIV have been shown to carry several elements, such as the trans-activation response element (TAR), pre-microRNA, and full-length genomic RNAs [9]. A further example of RNA transport enabled by exosomes involves the tick-borne Langat virus (LGTV), a flavivirus that shares similarities with tick-borne encephalitis virus (TBEV). This RNA virus stimulates tick cells to release exosomes, which are subsequently taken up by neuronal cells [10]. The observations made herein support the hypothesis that EVs are capable of transporting both small and large RNA molecules. It is likely that future research will identify extracellular viral lncRNAs that are significant in the context of pathogenesis. This could lead to the advancement of new and effective diagnostic and therapeutic approaches utilizing EVs.

As the field of small non-coding RNAs (sncRNAs) continues to expand, it has become evident that several species have not been thoroughly studied or have been neglected in relation to animal infectious diseases. Among these overlooked entities are piRNAs, small nucleolar RNAs (snoRNAs), and circular RNAs (circRNAs) [11]. Long non-coding RNAs can act as valuable biomarkers not only for diagnosing diseases but also for identifying species, assessing fertility in livestock, and evaluating the production potential of meat and eggs in poultry, and may also be explored as candidates for vaccines against emerging diseases in the future [12]. To effectively bridge the research gaps that currently exist, it is essential to establish lncRNA databases that serve both veterinary and human medical applications. A significant question pertains to the degree of conservation of the epigenomic and regulatory frameworks of these lncRNAs across animal species and how this conservation may relate to zoonotic potential or the cross-species transmission of pathogens [11]. The field encounters numerous challenges, from the discovery phase through validation and translation into clinical practice, with specific hurdles arising in veterinary contexts and across various omics platforms. Addressing these challenges in a thorough manner is essential for advancing the field. Future studies may aim to elucidate the role of long non-coding RNAs (lncRNAs) in the detection of parasitic infestations and bacterial loads in various animal species. The investigation of bacterial loads in food samples through the detection of lncRNAs in bacteria could provide valuable insights. Moreover, this research could be expanded to include interactions between bacteria and bacteriophages, particularly focusing on how lncRNAs are utilized by phages to effectively target and destroy bacterial populations.

This review provides an overview of long non-coding RNAs, detailing their classification, their involvement in the innate antiviral immune response, and their significance in viral pathogenesis, as well as various viral diseases affecting chickens. While comprehensive studies are still needed across all poultry diseases, this review highlights several viral diseases where research on lncRNAs has been conducted.

## 2. Long Non-Coding RNAs (lncRNAs)

Diseases caused by viruses pose a great challenge to human health, and their development has been driven by the imbalanced host immune response. Host innate immunity is an evolutionary defense system that is critical for elimination of the virus. An overactive innate immune response can also lead to inflammatory autoimmune diseases, which require precise control of innate antiviral response for maintaining immune homeostasis. Innate immunity is the first and most rapid line of defense against the invasion of microbial pathogens [13]. Non-coding RNAs with a length of at least 200 nucleotides are referred to as lncRNAs [14].

### 2.1. Classification of Long Non-Coding RNAs (lncRNAs)

In order to gain a deeper understanding of the functional roles of lncRNAs, it is beneficial to categorize them into distinct groups. This classification aids in investigating their underlying mechanisms of action, generating new hypotheses, and offering insights into the variations among the primary classes of lncRNAs. Conventional lncRNAs are divided into five classes based on their position in relation to protein-coding genes (P-CGs): (i) long intergenic transcripts; (ii) intronic lncRNAs (located within the intron of P-CGs); (iii) bidirectional lncRNAs (transcribed in opposite directions with the promoter of P-CGs); (iv) antisense lncRNAs (transcribed across the exons of a P-CGs from the opposite direction); and (v) pseudogene-type lncRNAs (transcribed from a gene without the ability to produce proteins) [15,16]. The classification of long non-coding RNAs (lncRNAs) is determined by their transcriptional origins within the genome. Intergenic lncRNAs are those that are transcribed from regions between genes, while intronic lncRNAs are those that are derived exclusively from the introns of protein-coding genes [17,18,19]. Thus, there are antisense, divergent, intron, intergenic, enhancer, promoter, and transcriptional start site-associated lncRNAs in the cluster of lncRNAs, depending on the relative positions of the lncRNA and the encoded gene [15]. Antisense lncRNAs are transcribed from the antisense strand of protein-coding genes, overlapping with either exonic or intronic regions, or they may span the entire protein-coding sequence through an intron. Research indicates that intergenic lncRNAs and intronic lncRNAs are likely governed by distinct transcriptional activation mechanisms. Furthermore, these two classes of lncRNAs may exhibit variations in poly(A) modifications and display functional activities in diverse cellular environments. However, the functional characterization of intronic lncRNAs is still limited, with only a small number of these transcripts having been thoroughly investigated. Long intergenic non-coding RNAs (lincRNAs) are involved in multiple functional mechanisms, including *cis* and *trans* transcriptional regulation (*cis*-acting lncRNAs play a crucial role in the regulation of gene expression for genes that are in close proximity within the genome, in contrast to *trans*-acting lncRNAs, which regulate the expression of genes that are more distantly located), translational control, splicing regulation, and other post-transcriptional regulatory activities. Additionally, there has been considerable research into the expression profiles and conservation of lincRNAs across different species [15]. It has been observed that lincRNAs undergo transcriptional activation in a manner similar to that of mRNAs. These molecules are more conserved than introns and antisense transcripts, exhibit a higher degree of tissue-specific expression compared to protein-coding genes, and are more stable than intronic lncRNAs. The “K4-K36” domain, which is indicative of active transcription in protein-coding genes due to histone H3K4 trimethylation at the 5′ end and histone H3K36 trimethylation throughout, is commonly found in transcriptionally active lincRNAs [20,21,22].

### 2.2. Biogenesis of lncRNAs

A significant proportion of lncRNAs are present in the nucleus, which leads to questioning the underlying factors that contribute to their varied localization. Comparative analysis of the global characteristics of lncRNAs and messenger RNAs (mRNAs) suggests that lncRNA genes are characterized by a lower degree of evolutionary conservation, a smaller number of exons, and reduced expression levels [23]. The reduced levels of long non-coding RNAs (lncRNAs) are likely linked to the existence of repressive histone modifications at their respective gene promoters. Furthermore, the transcriptional mechanisms of these lncRNAs may account for some of their distinctive properties. The phosphorylation status of the carboxy-terminal domain of RNA polymerase II (Pol II) reflects various stages of transcription, with a notable fraction of lncRNAs being transcribed by Pol II that exhibits dysregulated phosphorylation. Such lncRNAs are characterized by weak co-transcriptional splicing, and termination of transcription at these genes occurs without reliance on polyadenylation signals. This leads to a temporal buildup of lncRNAs on chromatin, which is then rapidly degraded by the RNA exosome [24]. These findings elucidate the rationale for the frequent localization of lncRNAs in the nucleus and suggest that, to achieve significant accumulation in particular cell types, they must successfully evade nuclear surveillance systems. In addition, lncRNAs typically possess embedded sequence motifs that can recruit specific nuclear factors. This recruitment is essential for nuclear localization and functioning of lncRNAs. Recent studies utilizing high-throughput massively parallel RNA assays (MPRNAs) have discovered a C-rich sequence from *Alu* repeats that promotes nuclear retention of lncRNAs by binding to the nuclear matrix protein, namely heterogeneous nuclear ribonucleoprotein K (hnRNPK) [25].

A significant proportion of lncRNAs are transported to the cytosol, suggesting that they likely utilize similar processing and export mechanisms as messenger RNAs (mRNAs). Once in the cytoplasm, lncRNAs are probably subjected to distinct sorting processes that direct them to particular organelles or facilitate their distribution throughout the cytoplasm, where they interact with various RNA-binding proteins (RBPs). Specific long non-coding RNAs (lncRNAs) exhibit associations with ribosomes, particularly those characterized by extended “pseudo” 5′ untranslated regions. These regions are termed “pseudo” due to their positioning before “pseudo-open reading frames” within the lncRNA structure. Degradation of these ribosome-associated lncRNAs may be initiated through a mechanism that is dependent on translation. However, it remains unclear whether these lncRNAs are actively utilized by ribosomes for translation, serve a functional role in the translation process, or simply exist passively within the ribosomal complex [4,26].

### 2.3. Mechanism of lncRNA Action

Protein interactors of lncRNAs, such as conventional and non-conventional RNA-binding proteins (RBPs), are critical to the achievement of lncRNA functions [8,9,10,11,12,13,16]. Many lncRNAs are gradually being recognized as key components of virus–host interactions, mainly through antiviral-response-independent and -dependent mechanisms [17,18]. lncRNAs can take part in X chromosome inactivation through interactions with various proteins [17], genomic imprinting [27,28,29,30], chromatin modifications [31,32,33,34], X chromosome inactivation via DNA methylation [35,36,37], and mRNA degradation [38,39]. lncRNAs also play the role of regulating the innate immune response either by adsorbing miRNA through the sponge effect or directly binding to the component of innate immune molecules [40]. Furthermore, they control mRNA splicing [41]. lncRNAs use transcriptional interference to control the splicing of genes [42], and some lncRNAs possess ORFs and use coding of one or more micro-peptides to demonstrate their unique regulatory roles [43].

After more than ten years of research, scientists have connected the molecular mechanisms of lncRNAs to significant regulatory molecules that are functional in a range of biological and pathological processes [44]. First, lncRNAs have the ability to control chromatin modifications. Rinn et al. [45] found that lncRNA HOTAIR interacts directly with polycomb repressive complex 2 (PRC2) and further modifies chromatin silencing mediated by PRC2. Besides regulating transcription, lncRNAs can also interact with proteins to control crucial signaling cascades and post-transcriptional processes. Additionally, lncRNAs can function as ceRNAs to “sponge” or absorb miRNAs and stop miRNAs’ inhibitory effect on the genes they target. Wang et al. [46], for instance, reported that lncRNA NRFs could bind hsa-miR-873, upregulate RIPK1/RIPK3, and enhance necrosis. Finally, it is possible that lncRNAs can influence the stability and translation of mRNA in the cytoplasm and play a role in controlling processes that are dependent on cells, including apoptosis, migration/invasion, and growth [47,48].

According to archetypal classifications, long non-coding RNAs (lncRNAs) exhibit five distinct molecular functions: they serve as signals, decoys, guides, scaffolds, and enhancers.

Signal lncRNAs are produced at particular times and locations within the cell as a reaction to various stimuli. Within this category, certain lncRNAs function as regulatory elements, whereas others are simply the incidental products of transcription processes, where initiation, elongation, or termination may play a regulatory role. Signal lncRNAs are recognized for their ability to engage with chromatin-modifying enzymes, including histone methyltransferases, to inhibit transcription of their target genes, either by obstructing the transcriptional process or by facilitating the formation of heterochromatin [33].

The term “decoy” lncRNAs refers to their role as molecular sinks reduce the accessibility of certain regulatory factors through their decoy binding sites. Repression of transcription by these lncRNAs occurs indirectly through sequestration of key regulatory components. This includes microRNAs (miRNAs), chromatin-modifying complexes, catalytic proteins, and transcription factors, which results in reduced availability of these essential factors. The presence of lncRNA decoys inhibits transcriptional activity by obstructing interaction between a designated effector and its corresponding intrinsic target. Some examples are TUG1, MEG3, H19, growth arrest-specific 5 (GAS5), HOTAIR, phosphatase and tensin homolog pseudogene 1 (PTENP1), and MALAT1 [33,49].

The organization and localization of factors at specific genomic loci are critically dependent on guide long non-coding RNAs (lncRNAs), which are integral to genome regulation. These lncRNA guides serve to direct chromatin modifiers and various protein complexes, including ribonucleoproteins (RNPs) and transcription factors, to targeted gene sites, facilitating their correct localization at the transcriptional loci. lncRNAs like HOTAIR, functional intergenic repeating RNA element (FIRRE), COOLAIR, KCNQ1 overlapping transcript 1 (KCNQ1OT1), taurine-upregulated gene 1 (TUG1), HOTTIP, XIST, MEG3, and ANRIL play a pivotal role as guides by altering chromatin structures and regulating the recruitment of epigenetic modifiers to their designated loci. The specific targets of guide long non-coding RNAs (lncRNAs) are activated by interactions between RNA and RNA, RNA and protein, and RNA and DNA, but the exact nature of this mechanism is still not well defined [50].

Scaffold lncRNAs are essential for the structural organization of multi-protein complexes, including those that are short-lived, such as ribonucleoprotein (RNP) complexes. Once these RNP complexes are fully assembled, they possess the ability to either inhibit or promote transcription, contingent upon the specific RNAs and proteins that are present and their characteristics. The lncRNA TERC serves as a notable example of a molecular scaffold which facilitates assembly of the telomerase complex by integrating associated proteins with reverse transcriptase activity within a singular ribonucleoprotein (RNP) structure [51,52].

The enhancer archetype illustrates how the interaction between chromatin and DNA is affected by enhancer regions (ERs), as mediated by enhancer RNAs (eRNAs). It is believed that these lncRNAs do not dissociate from ERs; rather, they function to bind proteins to these enhancer sites [53].

### 2.4. Role of lncRNAs in Gene Expression

RNA polymerase II (Pol II) is responsible for the transcription of lncRNAs, and they share a similar biogenesis with mRNAs due to their polyadenylation and 5′-cap formation. Although their exon count and splicing efficiency are typically lower than those of mRNAs, lncRNAs frequently undergo splicing [54,55,56]. lncRNAs control gene expression on a number of levels. lncRNAs can influence chromatin structure and function, the transcription of nearby and distant genes, RNA splicing, stability, and translation, and more by interacting with proteins, DNA, and RNA. Moreover, lncRNAs have a role in the synthesis and control of nuclear condensates and organelles.

#### 2.4.1. Chromatin Structure Regulation

Detection of RNA–chromatin associations and chromatin conformation capture techniques reveal complex regulation of chromatin architecture and gene expression by lncRNAs. RNA has inherent regulatory potential, as its negative charge can neutralize histone tails, leading to chromatin de-compaction [57,58]. Nuclear lncRNAs interact with DNA to alter the chromatin environment, either indirectly through affinity for proteins or by binding to specific DNA sequences. Protein-assisted long-range chromatin interactions can facilitate direct lncRNA transcriptional effects on target genes [59]. According to Tan-Wong et al. [60] and Niehrs et al. [61], lncRNAs can create hybrid structures with DNA, influencing chromatin accessibility. These interactions can take the form of triple helices or R-loops, which can pose risk to genome stability. However, recent findings suggests that this can be an effective regulator of gene expression and coordinator of DNA repair, with lncRNAs interacting with these structures [37,62,63,64,65].

#### 2.4.2. Regulation of Transcription by lncRNA

One of the main factors influencing the regulatory interaction between a lncRNA and its neighboring genes is their relative position. Given that bidirectional and extensive antisense lncRNA transcription has been shown to be evolutionarily conserved [66], the non-random genomic distribution of lncRNAs may indicate that genes have evolved to regulate their own expression in a context-specific way. For example, divergent lncRNAs’ genomic organization is essential for *cis*-gene regulation. According to Luo et al., two primary, non-exclusive methods can mediate this regulation: either the lncRNA transcript regulates neighboring loci, or the transcription or splicing process of the lncRNA results in a chromatin state or steric hindrance that affects the expression of neighboring genes [67]. By interfering with the transcription machinery, lncRNAs can decrease the expression of genes through modifying the recruitment of transcription factors or Pol II to the blocked promoter [42], altering histone modifications [68], and decreasing chromatin accessibility [69]. In 2019, Rom et al. [70] mentioned that the conserved lncRNA CHD2 adjacent, suppressive regulatory RNA (CHASERR), which is situated upstream of the chromatin remodeler Chd2 gene, exemplifies an additional way that lncRNAs can control extensive transcriptional inhibition. It was discovered that accessibility at the Chd2 promoter and numerous additional promoters—all of which were controlled by CHD2—was enhanced by CHASERR depletion. Membraneless RNA–protein compartments called nuclear condensates are essential for numerous biological activities. Various abundant lncRNAs have scaffolding or regulatory functions that are necessary for the building and operation of multiple nuclear condensates [71].

#### 2.4.3. Role of lncRNA in Post-Transcriptional Regulation

By binding to RNA sequence motifs or structures, proteins can be sequestered by lncRNAs and form specific lncRNA–protein complexes (lncRNPs). This results in altered mRNA splicing and turnover, as well as modulation of signaling pathways in certain biological contexts. lncRNAs regulate many other aspects of gene expression in addition to their involvement in nuclear organization and transcription regulation. Some lncRNAs are even translated into functional peptides [72]. *trans*-acting lncRNAs establish distinct structural motifs or engage in sequence-based interactions with RBPs. According to Yap et al., pyrimidine-rich non-coding transcript (PNCTR) inhibits PTBP1-mediated mRNA splicing elsewhere in the nucleoplasm by confining pyrimidine tract-binding protein 1 (PTBP1) to the perinucleolar compartment (PNC) [73]. Non-coding RNA triggered by DNA damage (NORAD), which is extensively produced in the cytoplasm after DNA damage, sequesters Pumilio (PUM) proteins to preserve genomic stability. PUM RBPs are sequestered in the cytoplasm by NORAD, which represses the translation and stability of the mRNAs it binds to [74,75,76]. Through base pairing, these *trans*-acting lncRNAs interface directly with other RNAs.

### 2.5. Role of lncRNAs in the Innate Antiviral Response

Proinflammatory activation and an antiviral response are brought on by the activity of host immune system, which plays a crucial role in the defense against viral infection, [77,78]. Upon viral infection, pathogen-associated molecular patterns (PAMPs) on viruses are recognized by pathogen recognition receptors (PRRs), including Toll-like receptors (TLRs), which in turn initiate the innate immune response [79,80]. According to Carnero et al., the lncRNA EGOT induced by TLR4/TLR7 amplifies viral replication and antagonizes the antiviral response [81]. Furthermore, the lncRNA EGOT may be involved in a number of processes during viral infection, given the critical roles played by TLR4 and TLR7 in host immunity and their crosstalk with transduction signaling pathways like the NF-κB and IFN pathways [82,83,84]. Further research is necessary to fully understand the precise downstream mechanisms of lncRNAs/TLRs in the antiviral immune response, even though the TLR-induced viral response causes upregulation of lncRNAs that modulate the innate immunity.

According to Doyle et al., interferons are the primary immunomodulatory and pro-inflammatory cytokines in the antiviral immune response, and they also aid in induction of lncRNA activity [85]. An example can be taken from the work of Kotzin et al., who established that, early on in both acute and chronic lymphocytic choriomeningitis virus infections, T cell receptor (TCR) and type I IFN activation induce the lncRNA known as Morrbid. The lncRNA Morrbid works by suppressing the PI3K/AKT pathway and inducing the pro-apoptotic gene BCL2L11, which in turn helps to regulate CD8 T cell survival and differentiation, thus helping in the immune response [86].

Not only are lncRNAs stimulated by the viral immune system, but they can also modulate the immune response by regulating important immune molecules like cytokines. According to Li et al., TNFα and hnRNPL related immune-regulatory lincRNA (THRIL) is necessary for TNFα expression. THRIL regulates TNFα downstream targets and interacts with hnRNPL directly. THRIL induces other cytokines and chemokines, including IL-8, CXCL10, and CSF1, but further research is needed to determine the mechanisms [87].

According to newly available research [88,89], lncRNAs have the ability to transcriptionally control gene expression, which can alter viral replication and the immune response. First, the transcription of innate immune genes like IFN and ISGs can be activated or repressed by lncRNAs through the recruitment of transcription factors (TFs) [90]. It was demonstrated by Ma and colleagues that the Hantaan virus (HTNV) induces the lncRNA NEAT1. The SFPQ is relocated to paraspeckles by induced NEAT1, which also reverses the transcriptional inhibition of RIG-I and DDX60. The IFN response induced by RIG-I is facilitated, and endogenous RIG-I expression is promoted, by restored DDX60. As a result, NEAT1 has the ability to influence the innate immune response and activate IFN signaling, providing negative feedback against HTNV viral infection [91].

dsRNA-binding proteins are important components of the antiviral innate immune system and play a crucial role in repressing viral replication by causing various changes in cellular and viral RNA processes [92]. Exogenous circRNAs can initiate an innate immune response that provides protection against viral infection. The authors found that, in various cell lines, circRNAs potently induced the expression of multiple innate immune system regulatory genes, such as retinoic acid-inducible gene-I (RIG-I), protein kinase R (PKR), melanoma-differentiation-associated gene 5 (MDA5), 2′-5′ oligoadenylate synthase 1 (OAS1), and OAS-like protein (OASL) [93].

### 2.6. Role of lncRNAs in Virus Pathogenesis

Increasing research works indicate that the expression of long non-coding RNAs (lncRNAs) plays a significant role in the regulation of numerous transcription factors (TFs) and proteins, which is crucial for both the replication of viruses and the activation of viral latency. Various studies have demonstrated that the expression of cellular lncRNAs can be modulated by viral infections. Additionally, certain viruses utilize their own encoded lncRNAs to enhance viral replication or maintain latency. Given how pleiotropic their functions are, it is not surprising that lncRNAs could play a role in virus replication. It has been discovered that viruses with low coding capacity control the expression and function of both host and viral genes by means of cellular lncRNAs. Various animal viruses have been demonstrated to dysregulate host lncRNA expression, including avian leukemia virus [94], herpes simplex virus [95], Marek’s disease virus [96], human immunodeficiency virus (HIV) [97], hepatitis B virus (HBV) [14], and severe acute respiratory syndrome coronavirus (SARS-CoV) [98]. Subsequent research by Yao et al. has demonstrated that viral genes control the level of cellular lncRNAs, which in turn controls the expression of genes encoding proteins and ultimately promotes viral infection [99]. The first viral lncRNA to be characterized is viroid. It consists of a circular, single-stranded RNA genome that does not have protein-coding potential. However, the viroid is able to replicate on its own [100].

#### 2.6.1. lncRNAs in Viral Gene Expression

The viral genome controls the expression of “late” proteins, which are required to assemble the capsid and package the viral genome, and “early” proteins, which allow genome replication once the virus enters the proper cell compartment. Moon et al. [101] reported that sub-genomic flavivirus RNA (sfRNA), a lncRNA partially degraded from viral genomic RNA, most likely by the cellular 5′–3′ exoribonuclease XRN1, inhibits XRN1 activity and modifies host mRNA stability in cells infected with dengue virus or Kunjin virus. This effect might interfere with the control of host cell gene expression and help stabilize viral transcripts. According to a study by Rossetto et al., polyadenylated nuclear (PAN) RNA, a lncRNA encoded by the genome of Kaposi’s sarcoma-associated herpesvirus (KSHV), can physically interact with the KSHV genome to transcriptionally activate expression of the KSHV gene [102]. Another study conducted in 2013 by Juranic Lisnic and colleagues demonstrated that PAN RNA can alleviate gene suppression by binding to the host poly(A)-binding protein C1 (PABPC1) to regulate mRNA stability and translation efficiency. This serves as a molecular scaffold for chromatin-modifying enzymes to remove the H3K27me3 mark, which is necessary for the production of late viral proteins [103].

#### 2.6.2. lncRNAs in Viral Replication

RNA–RNA interactions and RNA–protein interactions are two ways that virus-encoded lncRNAs control viral replication. Numerous secondary stem-loop II structures can be found in the 3′ untranslated region (UTR) of the flavivirus RNA genome. These structures prevent the nuclease XRN1 from breaking them down, which produces the functional lncRNA, i.e., sfRNA [104]. It was reported that regulation of the efficiency of flavivirus genome replication was significantly influenced by sfRNA. The host’s innate immune response has the ability to produce miRNA and eliminate viral genomic RNA during a flavivirus infection. But in order to protect viral genomic RNA, sfRNA bound to the miRNAs and caused their degradation, which allowed the virus to replicate in the host cell [105]. Once the newly generated viral genomes and proteins are adequately prepared within the host cells, they are packaged, assembled, and released. Additionally, these processes involve HBoV1 BocaSR viral lncRNAs [106]. Whereas RNA viruses copy their genomes directly to RNA, DNA viruses copy their genomes straight to DNA. On the other hand, certain RNA viruses and certain DNA viruses use DNA and RNA intermediates to copy their genomes [107].

#### 2.6.3. lncRNAs in Viral Assembly and Release

According to findings, Japanese encephalitis virus (JEV) sfRNA is a trans-acting riboswitch that promotes genomic RNA synthesis, packaging, and virion release while inhibiting the translation of JAV and host antigenomic genes. It is found in the late stages of the viral replication cycle [105]. Thus, these results indicate that lncRNAs produced by viruses are probably involved in controlling the entire viral life cycle, which includes viral genome replication, gene expression, assembly, and release of virions.

Some viruses participate in a lysogenic cycle characterized by the incorporation of their genome into a specific region of the host’s chromosome through genetic recombination. As a result, replication of the viral genome occurs concurrently with division of the host cell. While many viral proteins are translated and induced to maintain persistent infections by directly inhibiting T cell responses and/or downregulating antigen recognition molecules, specific lncRNAs have been demonstrated to interfere with the immune response, allowing for the continued presence of the virus [33].

### 2.7. Methods of Detecting lncRNAs

Based on sequence similarity and the existence of long open reading frames (ORFs), it is possible to anticipate protein-coding gene transcripts and several families of non-coding RNAs (tRNAs and rRNAs) with a fair degree of accuracy, as mentioned in work by Ilott and Ponting from 2013 [108]. The following is a list of some of the techniques.

#### 2.7.1. Full-Length cDNA Sequencing

According to Carninci et al. [109], creating a whole cDNA sequence is the best way to create precise transcript models. The Functional Annotation of The Mammalian Genome project (FANTOM) was the first effort to characterize the entire coding capacity of the mammalian genome. Cap-analysis of gene expression (CAGE) tag sequences from FANTOM3 was used to identify transcriptional start sites (TSSs) for future FANTOM data releases. Using this technique, approximately 30% of all transcripts analyzed were categorized as non-coding RNAs.

#### 2.7.2. Chromatin State Maps

Histone alterations are conserved among mammalian species, and their associations with transcriptional activity justify their use as indicators of various regulatory components [110]. Particularly, extensive investigations using chromatin immunoprecipitation in conjunction with hybridization to microarray tiling arrays (ChIP-chip) have demonstrated that certain histone marks are suggestive of particular regulatory elements [110,111]. Guttman and colleagues found around 1250 potential lncRNA loci in various mouse cell lines using the K4–K36 mark. After analyzing expression data from four mouse cell lines, embryonic development over time, and multiple adult tissues, the researchers concluded that these identified lncRNAs may play a role in various biological processes, such as ESC pluripotency, neuronal development, and immune function. This study successfully uncovered numerous lncRNAs by leveraging knowledge of the connection between chromatin state and transcriptional activity, although it had some limitations [20].

#### 2.7.3. RNA Sequencing

RNA-seq deep sequencing provides a more straightforward method for evaluating transcriptomes compared to analysis of chromatin markers. By sequencing short cDNA fragments, it is possible to effectively reconstruct the entire transcriptome, delivering a broader dynamic range than microarrays. This technique also enables the identification of novel loci and transcripts, in addition to quantifying the prevalence of alternatively spliced variants [112]. The lack of evidence for transcripts that extend across adjacent protein-coding genes and lncRNA loci serves as an additional criterion for differentiating the two. Consequently, researchers aiming to identify functional lncRNAs within their specific systems consider RNA-seq to be highly beneficial, as mentioned by Ilott and Ponting [108] and shown in Figure 1.

Subsequent computer reconstruction of transcriptomes has made it possible to identify a number of lncRNA primary characteristics. For instance, the locations of lncRNA loci and protein-coding genes are not always the same; their transcripts can be in intergenic sequence, within the introns of protein-coding genes, or overlapped on the same or opposite strand. Only RNA-seq (or cDNA sequencing) can provide the possibility of classifying lncRNAs other than intergenic non-coding RNAs. This observation was based on research conducted in 2012 by Pauli and colleagues [113] using RNA-seq during embryogenesis to identify long non-coding RNAs expressed in zebrafish embryos in a systematic manner. They identified 1133 non-coding multi-exonic transcripts expressed during development. These consist of precursors for small RNAs (sRNAs), intronic overlapping lncRNAs, exonic antisense overlapping lncRNAs, and long intergenic ncRNAs (lincRNAs). They noticed that the temporal expression profile of lncRNAs disclosed two novel characteristics: they were selectively enriched in early-stage embryos and were expressed within smaller time windows compared to protein-coding genes. Moreover, a number of lncRNAs exhibited unique subcellular localization patterns and tissue-specific expression.

#### 2.7.4. Chromatin Isolation by RNA Purification (ChIRP)

The ChIRP method, developed by Tian et al. in 2020 [114], enabled high-resolution genome-wide mapping of lncRNA. This technique has successfully identified a significant number of lncRNAs, along with their chromatin interaction complexes. Furthermore, there is a growing interest among researchers in mapping individual lncRNAs to their related genomic loci across the entire genome. The fundamental steps involved in the ChIRP technique include (1) design and synthesis of biotinylated anti-sense DNA tiling probes; (2) crosslinking and sonication of the samples; (3) hybridization of biotinylated DNA probes to RNA, leading to the isolation of the bound chromatin; and (4) purification of RNA, DNA, or protein from the ChIRP samples, which is then subjected to omics-based analysis. ChIRP has emerged as a remarkably efficient and powerful instrument for conducting functional analyses of long non-coding RNAs (lncRNAs), thus serving as an important method for elucidating the roles of lncRNAs in various biological contexts, including development, inflammatory processes, and cancer. Research has shown that lncRNAs are integral to the regulation of developmental genes in a variety of processes, including erythropoiesis, spermatogenesis, and chondrogenesis. Through the ChIRP assay, researchers have identified the multifaceted regulatory roles of lncRNAs in developmental biology [115].

#### 2.7.5. Modified Crosslinking and Immunoprecipitation (M-CLIP) Assay

The interaction of competing endogenous RNAs (ceRNAs) with microRNAs (miRNAs) is particularly crucial in the field of cancer biology. Such interactions lead to the sequestration of miRNAs, thereby obstructing their binding to mRNA targets and consequently altering gene expression patterns that are essential for cancer development and metastasis. Gaining insight into these interactions is critical for unraveling the complex regulatory frameworks that underline disease pathology. In traditional CLIP assays, RNA is crosslinked to protein using UV or chemical methods, and the RNA–protein complex is then immunoprecipitated to examine the RNA component and detect regulatory and functional mechanisms. These techniques are useful for researching RNA–protein interactions, but they are insufficient for focusing on lncRNA–miRNA interactions. In order to overcome this limitation, researchers have adapted the CLIP methodology to concentrate on interactions between lncRNA and miRNA within ribonucleoprotein complexes that include Argonaute 2 (Ago2). Ago2 is a crucial protein in the RNA-induced silencing complex (RISC), which plays a significant role in the regulation of miRNA activity and stability. By integrating Ago2 antibodies into the immunoprecipitation process, it becomes possible to selectively examine the Ago2 immune complexes that encompass both lncRNAs and the miRNAs they bind. This method has inherent limitations, as it is designed to identify interactions rather than provide an accurate quantification of the number of miRNA molecules sequestered by lncRNAs or determine their precise binding affinities. Moreover, the method primarily concentrates on interactions with recognized proteins, such as Ago2, and may overlook those involving unidentified or novel proteins unless they are explicitly targeted, thereby potentially restricting the identification of new interaction partners [116].

#### 2.7.6. Single-Cell RNA Sequencing

The advent of single-cell RNA sequencing has enabled analysis of the transcriptome at the individual cell level for millions of cells within a single study. This technological advancement facilitates the classification, characterization, and differentiation of each cell based on its transcriptomic profile, thereby allowing for the identification of rare yet functionally significant cell populations [117]. Qu et al. conducted single-cell RNA sequencing (scRNA-seq) on peripheral blood mononuclear cells (PBMCs) derived from both Avian leukosis virus subgroup J (ALV-J)-infected chickens and control subjects at 21 days post-infection (DPI). The objective was to elucidate subsets of chicken PBMCs, along with their distinct molecular and cellular attributes. The results presented a valuable repository of gene expression profiles for chicken PBMC subsets, enhancing systems-level understanding of their functions under homeostatic conditions and in response to viral infections. The findings revealed that ALV-J infection significantly influenced the composition of PBMCs when compared to control groups. B cells demonstrated a limited response, with a decrease in their numbers noted in PBMCs from chickens infected with ALV-J. In contrast, the proportions of cytotoxic Th1-like cells and CD8+ T cells increased within the T cell population of PBMCs from ALV-J-infected chickens, highlighting their potential role as crucial effectors in the response to ALV-J infection [118]. A primary limitation of single-cell RNA sequencing lies in the substantial noise present within its datasets, which is largely due to the limited quantity of starting materials. This noise can lead to inaccurate downstream analyses and misleading results. Another challenge is that analysis of single-cell RNA sequencing (SC-RNA-seq) data requires the development of cutting-edge computational strategies, algorithms, and tools to ensure the effective performance of data processing tasks such as normalization, differential gene expression (DGE) analysis, cell imputation, and dimensionality reduction [119].

## 3. Role of lncRNAs in Specific Viral Chicken Diseases

Long non-coding RNAs have a significant impact on regulating gene expression in addition to genes, which are essential in determining the host immune response. Extended non-coding RNAs impact genes at various phases, such as chromatin remodeling-induced activation or inactivation, transcription activation or suppression through binding to transcription factors, translation inhibition and splicing modulation through transcript binding, and mRNA degradation through microRNA regulation. It has also been reported by Statello et al. that long non-coding RNAs have an impact at the protein level [4].

### 3.1. Avian Leukosis

Infection with avian leukosis virus J (ALV-J) causes hemangioma and hematopoietic malignancy in myeloid leukemia in chickens. On the other hand, nothing is known about the mechanisms underlying the distinct pathophysiology of ALV-J. Research on differentially expressed lncRNAs (DElncRNAs) between non-infected and infected tissues [120,121,122], in chick embryo fibroblast cells [123], or in primary monocyte-derived macrophages from chickens [124] showed that they have the potential to interact with immune-related miRNAs and genes, demonstrating the role of DElncRNAs in disease processes. According to Dai et al., a number of lncRNAs (XLOC_672329, ALDBGALG0000001429, XLOC_016500, and ALDBGALG0000000253) were hypothesized to cis-regulate cholesterol 25-hydroxylase [(CH25H)/cytokine-inducible SH2 containing protein (CISH)/interleukin 1 beta (IL-1β)/CD80 molecule (CD80)] in order to take part in host antiviral responses [124].

Hu and colleagues utilized RNA-seq to conduct a systematic review of lncRNAs in CEF cells and analyzed the gene expression profiles of ALV-J-infected chicken cells. Target prediction analysis showed that five lncRNAs––MG066601, MG066617, MG066602, MG066618, and MG066603––may act in *cis* or *trans* and affect the expression of genes involved in anti-viral innate immune responses. They found 36 differentially expressed lncRNAs and 91 genes, and the results showed that the JAK-STAT signaling pathway, toll-like receptor, RIG-I receptor, and NOD-like receptor were all enriched [123].

### 3.2. Marek’s Disease

MDV is a member of the α-Herpesviridae family of herpesviruses, with linear dsDNA and which causes tissue hyperplasia and neoplasia, along with a highly contagious phymatosis. The pathophysiological mechanism of MD is prompted by proteins encoded by the virulent genes of MDV. There are three serotypes of MDV: namely, serum type 1 (MDV-1) or Gallid herpesvirus (GaHV-2), serum type 2 (MDV-2) or Gallid herpesvirus 3 (GaHV-3), and serum type 3 (MDV-3), which is also known as Meleagrid herpesvirus I (MeHV-1) [125]. Three different types of lncRNAs—ERL lncRNA, linc-GALMD3, and linc-stab1—are involved in the MDV infectivity process.

Figueroa et al., in 2016 [125] reported that the ERL lncRNA is a naturally occurring antisense transcript of the MDV carcinogen, with a length of 7.5 kbp. It is connected to the adenosine to inosine acting on RNA (ADAR1) protein. During the lytic and latent phases of viral infection and reactivation, ERL lncRNA is expressed. In chickens, the JAK/STAT (Janus kinase/signal transducer and activator of transcription) and IFN-response pathway controls the expression of ADAR1 via an inducible promoter containing IFN-stimulated response elements [126]. Phylogenetic analysis reveals that chicken ADAR1 is closely related to the protein found in amphibians, and preliminary evidence suggests that it may play a role in the editing of viral RNA. Interferon-sensitive expression of ADAR1 is activated via the JAK/STAT signaling cascade. This activation triggers the phosphorylation of both STAT-1 and STAT-2, which then associate with interferon regulatory factor 9 (IRF-9) to interact with the interferon-stimulated response element (ISRE) found in the promoters of ISGs. Regulation of this pathway is mediated by suppressor of cytokine signaling 1 (SOCS1), which serves to prevent phosphorylation of STAT-1 and STAT-2 [127].

According to research by Burnside et al. [127] and Zhao et al. [128], the unedited ERL lncRNA has a long half-life because stable introns have been removed, and expression of most of the miRNA MDV-miR-M4 and lytic-associated miRNA MDV-miR-M1 has been inhibited.

According to work done by Zhang and colleagues in 2021, gga-miR-223 is a downstream non-coding RNA whose expression is regulated by the long intergenic non-coding RNA linc-GALMD3. Expression of the gga-miR-223 gene was markedly reduced by lincGALMD3 knockdown in MDCC-MSB1 cells, while the expression of other genes was increased [129]. According to Han and colleagues [130], linc-GALMD3 *trans*-regulates the expression of other genes in the chicken genome and cis-regulates expression of gga-miR-223. Using RNA-seq and qRT-PCR, they discovered and confirmed that MDV-infected CD4+ T cells had high expression of linc-GALMD3. After linc-GALMD3 function was lost by shRNA, RNA-seq analysis in MDCC-MSB1 cells revealed that linc-GALMD3 could positively *cis*-regulate expression of the downstream gga-miR-223 gene. Although there is not any experimental support for gga-miR-223, it targets the insulin-like growth factor 1 receptor (IGF1R), which controls MD lymphoma [131,132].

A particular kind of lincRNA called linc-SATB1 controls the MD resistance gene SATB1. SATB1 functions as a genome organizer that modulates chromatin structure and acts as a transcription factor for numerous genes critical for T cell development and activation. Current research predominantly indicates that SATB1 is linked to malignant tumors, tumor progression, and carcinogenic processes, suggesting a potential connection between SATB1 and Marek’s disease [133]. Its expression profile, when compared to other protein-coding genes, indicated a positive correlation with immune-related processes such as defense response, inflammatory response, and lymphocyte activation, as well as responses to external stimuli. In contrast, linc-SATB1 showed a negative correlation with cell cycle-related activities, including the cell cycle process and DNA replication. Furthermore, linc-SATB1 was notably expressed in infected birds of the MD-resistant line 6_3_ at 10 days post-infection (dpi), coinciding with the latent phase of MDV infection. These observations imply that linc-satb1 may play a significant role in the immune response to MDV. In order to gain complete insight into the involvement of linc-SATB1 in Marek’s disease and its effects on immunity, it is essential to conduct further studies utilizing specialized experimental methods, including RNA immunoprecipitation (RIP) and overexpression. The lincRNAs associated with Marek’s disease exhibit characteristics similar to those identified in mammalian genomes. These characteristics include notably reduced expression levels, abbreviated transcript lengths, a diminished number of exons, and lower sequence conservation relative to protein-coding genes [134].

lncRNA may interact with both host MDV-susceptible and -resistant genes to actually participate in MDV replication. According to Bacon [1], model animals for the investigation of responsible biomarkers or clinical diagnosis during MDV infection are chicken MD-resistant line 6_3_ and chicken MD-susceptible line 7_2_. Chickens with and without MDV infection had linc-GALMD1, a DElincRNA. The expression of immunoglobulin lambda-like polypeptide 1 (IGLL1) varies from line 6_3_ to line 7_2_. Following MDV infection, it expresses more in line 7_2_ chickens than in line 6_3_ chickens, suggesting that IGLL1 may be a line-specific or susceptible gene in response to MD [134].

### 3.3. Infectious Bursal Disease (Gumboro Disease)

Within the family Birnaviridae, the non-enveloped double-stranded RNA virus known as infectious bursal disease virus (IBDV) can cause acute, highly contagious, and immunosuppressive disease in chickens at as early as 3–6 weeks of age. This can result in a significant increase in mortality, as well as economic losses. The profound immunosuppression of broilers and egg-laying hens, as well as their heightened susceptibility to other illnesses and vaccination failure, are the primary causes of indirect losses in Gumboro disease [135].

Studies have been conducted to analyze the effect the of lncRNAs on IBDV infection, during which the host innate immune system produces dendritic cells (DCs), which play a unique role in both the innate and acquired immune systems during virus infection [135]. A microarray study conducted by Lin et al. [136] on DCs stimulated with IBDV and non-stimulated DCs showed that 965 mRNAs, 18 miRNAs, and 114 lncRNAs were expressed differently after IBDV stimulation. They also came to the conclusion that functional annotation of DElncRNA genes revealed relationships with the RNA biosynthesis process, protein localization, cellular response to starvation, and other concepts. According to pathway analysis, these were involved in the JAK-STAT/MAPK/mTOR/neurotrophin/CCR5/Interleukin-17 (IL-17) signaling pathways. Additionally, it was found that the target genes of miRNAs influence not only the MAPK signaling pathway but may also be involved in the regulation of protein localization and transport, which could impede viral migration and replication. Earlier research has shown that, in the context of IBDV infection, regulatory factors such as EIF2AK2, MX, GBP7, and IFIT can initiate the IFIT5-IRF1/3-RSAD5 signaling pathway in DF-1 cells, potentially restricting viral replication during the initial phase of infection.

lncRNA loc107051710 plays a crucial role in the antiviral immune response by modulating the production of type I interferon, STAT proteins, and interferon-stimulated genes, thereby inhibiting IBDV infection in chickens [137]. Recent studies indicate that downregulation of loc107051710 leads to increased replication of IBDV, attributed to a decrease in interferon production. A study on the antiviral activity of long non-coding RNA loc107051710 during infectious bursal disease virus infection owing to increased interferon production was conducted by Huang and colleagues in 2019 [137]. Following investigation of the relationship among loc107051710 and IRF8, type I IFNs, STATs, and ISGs, IBDV infection was induced in cultured DF-1 cells. RNA-seq was used to analyze the expression of mRNAs and lncRNAs in order to determine the antiviral activity of loc107051710. The findings demonstrated that loc107051710 positively controls IRF8 expression. Furthermore, the researchers discovered that loc107051710 functions as a positive transcriptional regulator of the antiviral genes Mx1, PKR, STAT1, STAT2, IFN-α, and IFN-β (Figure 2). FISH analysis revealed that the quantity of loc107051710 increased and transitioned from the nucleus to the cytoplasm during the course of infection. This observation suggests that loc107051710 is involved in antiviral mechanisms that operate not only at the transcriptional level but also at the post-transcriptional level.

### 3.4. Avian Influenza

Over the past ten years, there has been a notable surge in lncRNA research, leading to exploration of the critical roles played by lncRNAs in the regulation of influenza virus infection. Few lncRNAs have been found to interact with viral components, while the majority of lncRNAs have been found to regulate type I IFN signaling and IFN-stimulated genes (ISGs) [13]. An example is the identification and analysis of lncRNAs in ducks (*Anas platyrhynchos*) in response to H5N1 influenza viruses, which was performed by Lu et al. in 2019 [138]. After examining the characteristics of 62,447 lncRNAs from zebrafish, chicken, human, mouse, and *C. elegans*, the researchers created a pipeline to identify lncRNAs by incorporating the features with transcriptomic data. They annotated 4094 duck lncRNAs using 151,970 assembled transcripts from RNA-seq data from 21 individuals across three tissues. The results indicated that 619 lncRNAs and 3586 (87.6%) lncRNAs located in intergenic regions had differential expression in H5N1-infected ducks when compared to duck protein-coding transcripts. The researchers verified that eight lncRNAs, following H5N1 virus infection, exhibited remarkably different expression in vitro (in duck embryo fibroblast cells, DEF cells) and in vivo (in duck individuals), suggesting that lncRNAs may have significant roles in the antiviral immune response to influenza A virus infection.

By blocking various stages of the host immune response, lncRNAs can also function as positive regulators of viral infection, according to the research performed by Ma et al. in 2016 [139]. One study by Ouyang and colleagues in 2014 [140] demonstrated how lncRNA NRAV is significantly downregulated during IAV infection. NRAV likely inhibits the initial transcription of various significant ISGs, such as MxA and IFITM3, by altering the histone modifications H3K4me3 and H3K27me3 in these genes. This, in turn, can promote IAV replication (Figure 3). A few such examples, with lncRNA regulating influenza virus replication, are mentioned in Table 1 [13,141,142].

### 3.5. Infectious Bronchitis

The avian infectious bronchitis virus (IBV), a member of the Gamma-coronavirus genus, affects the kidneys, reproductive system, and upper respiratory tract of chickens [142].

According to research published in 2019 by An and co-workers [149], the coronavirus infectious bronchitis virus (IBV) causes virus-infected cells to produce a unique non-coding RNA (ncRNA). With the exception of a poly(A) tail from the 3′ untranslated region of the IBV genome and a 63 nt terminal leader sequence from the 5′ end of the viral genome, this ncRNA was composed of 563 nucleotides. This non-coding RNA was discovered to be a sub-genomic RNA produced by a discontinuous transcription mechanism, and it was revealed through the use of mutagenesis and reverse genetics techniques. In the IBV-infected cells, they discovered the presence of a novel sgRNA that is primarily derived from the 3′ UTR of IBV. A truncated core sequence, 27104UAACA27108, which was identical to nucleotides 3–7 of the IBV core sequence, facilitated synthesis of this sgRNA, which contributes at least three nucleotides (A27105, A27106, and C27107) to the efficient production of sgRNAs, further supporting the idea that non-canonical transcriptional signals are employed in the synthesis of coronavirus sgRNAs. Using mRNA, miRNA, and lncRNA microarray analysis, Lin et al. in 2019 [150] investigated host responses against IBV. The study offered details on the molecular pathophysiology and interactions between the virus and the host, avian bone marrow-derived dendritic cells (BMDCs).

### 3.6. Newcastle Disease

Differentially expressed genes and lncRNAs during NDV challenge were discovered by Vanamamalai and colleagues in 2023 [151], and these findings may have some bearing on the observed differential resistance pattern. It was discovered that there were 1580 lncRNAs and 552 genes with differential expression. In total, 52 annotated genes and a greater number of positively correlating lncRNAs were found to be downregulated, according to pathway analysis and gene ontology.

Sha et al. in 2024 [152] utilized RNA sequencing to explore the transcriptional profiles of visceral tissues of chicken embryos (CEVTs) subjected to infection by either the virulent NA-1 strain or the avirulent LaSota strain, assessing samples at 24 and 36 h post-infection. The analysis of long non-coding RNAs (lncRNAs) indicated that the significant pathological alterations and clinical manifestations resulting from virulent Newcastle disease virus (NDV) infection could be, in part, linked to associated target genes that are modulated by differentially expressed lncRNAs, including MSTRG.1545.5, MSTRG.14601.6, MSTRG.7150.1, and MSTRG.4481.1 (Figure 4). The findings reveal that virulent NDV infection capitalizes on the host’s metabolic resources and modifies the host’s metabolic functions, which is associated with heightened activation of the immune response.

## 4. Conclusions

This review focused on several dimensions of long non-coding RNAs (lncRNAs), detailing their classification, functional roles, mechanisms of action, and potential implications for antiviral innate immunity in chickens. The discussion highlighted the diverse roles of lncRNAs, including their functions as miRNA sponges, guides, scaffolds, and decoys, with specific examples. The abovementioned discussion on lncRNAs in viral chicken diseases has provided an assessment of the current state of knowledge regarding their locations, functions, and mechanisms. One of the principal challenges in the study of lncRNAs is insufficient accuracy in detecting transcripts, which entails numerous data filtration steps. Recent research utilizing transcriptomic methodologies has uncovered the involvement of lncRNAs and circular RNAs (circRNAs) in the regulation of myofibers in chickens. lncRNAs have a lot of potential for usage as therapeutic targets and indicators to achieve an antiviral state because of their proviral and antiviral properties. Host lncRNAs are considered to have a role in protecting the host from viral invasion. On the other hand, viruses utilize some lncRNAs to infect host cells and replicate themselves. These findings pave the way as a diagnostic tool to assess viral disease progression in a flock. Moreover, the role of lncRNAs in modulating the host’s innate immune response to viral infections, being used as biomarkers for species identification (by assessing the specific role of lncRNA in different species), or studying the role of lncRNA in myofibers and egg production in chickens may lead to future research toward developing genome editing tools involving chromatin isolation by RNA purification (ChIRP-Seq), ribosome profiling, RNA structure mapping, crosslinking immunoprecipitation (CLIP), CRISPR, oral vaccine candidates against tumorigenic and/or viral diseases, and also the development of algorithms to study zoonosis or cross-species transmission of diseases. Recently, research on non-coding RNAs has expanded, with increasing attention given to both lncRNAs and circRNAs. Future research could emphasize the significance of lncRNAs and circular RNAs in the detection of parasitic infestations or bacterial infection in animals. Furthermore, there is huge potential for expanding research to explore the dynamics of molecular interactions and expand the use of lncRNAs and circular RNAs as therapeutic targets [153].

## Figures and Tables

**Figure 1 ncrna-11-00042-f001:**
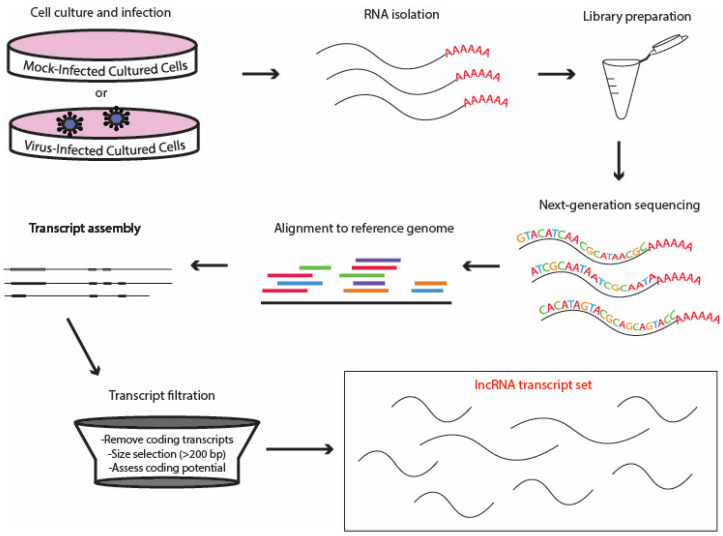
lncRNA identification workflow [107]. Workflow for identifying long non-coding RNA (lncRNA) transcripts from cultured cells. Starting with cell culture and infection (mock- or virus-infected), RNA is isolated, followed by library preparation and next-generation sequencing. Reads are aligned to a reference genome, and transcripts are assembled and filtered to remove coding transcripts, select those >200 bp, and assess coding potential, resulting in a set of lncRNA transcripts.

**Figure 2 ncrna-11-00042-f002:**
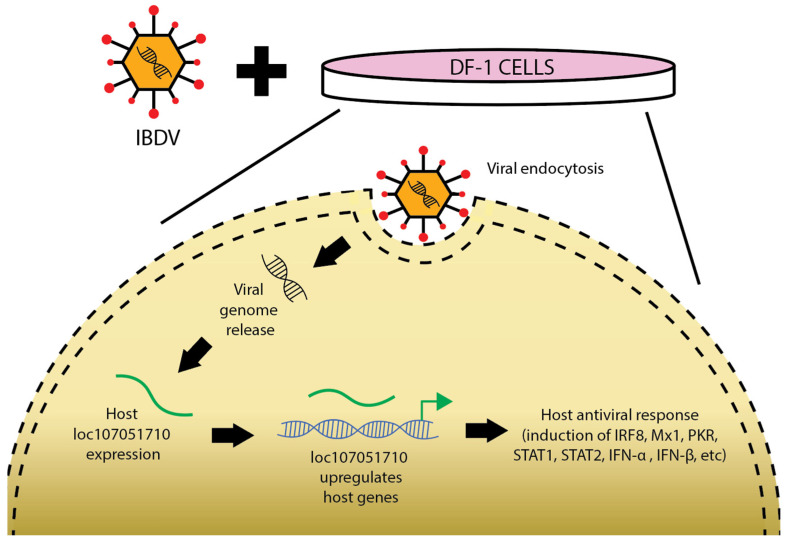
Action of IBDV-induced lncRNAs under in vitro conditions [137]. This diagram depicts the infection process of infectious bursal disease virus (IBDV) in DF-1 cells. The virus binds to the cell surface and enters through endocytosis. Inside the cell, the viral genome is released, leading to expression of the host gene loc107051710. This gene upregulates other host genes, triggering an antiviral response that includes the induction of IRF8, Mx1, IFN-γ, PKR, STAT1, STAT2, IFN-α, and IFN-β.

**Figure 3 ncrna-11-00042-f003:**
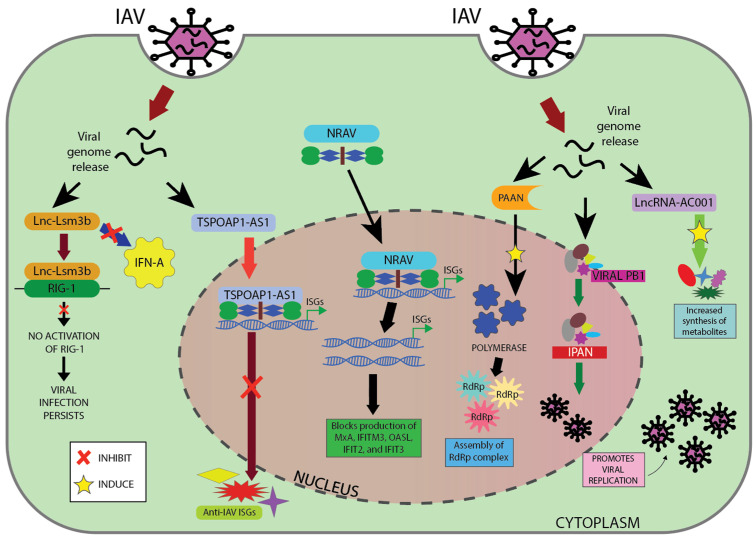
Mechanism of lncRNA acting as positive regulator for viral replication. This image shows how influenza A virus (IAV) releases its genome into a host cell and the different cellular responses. On the left, lnc-Lsm3b promotes viral infection by blocking RIG-I activation and IFN-α production. In the center, TSPOAP1-AS1 and NRAV block antiviral gene production by inhibition of ISG transcription. On the right, IPAN and PAAN help the virus replicate by aiding formation of the polymerase complex.

**Figure 4 ncrna-11-00042-f004:**
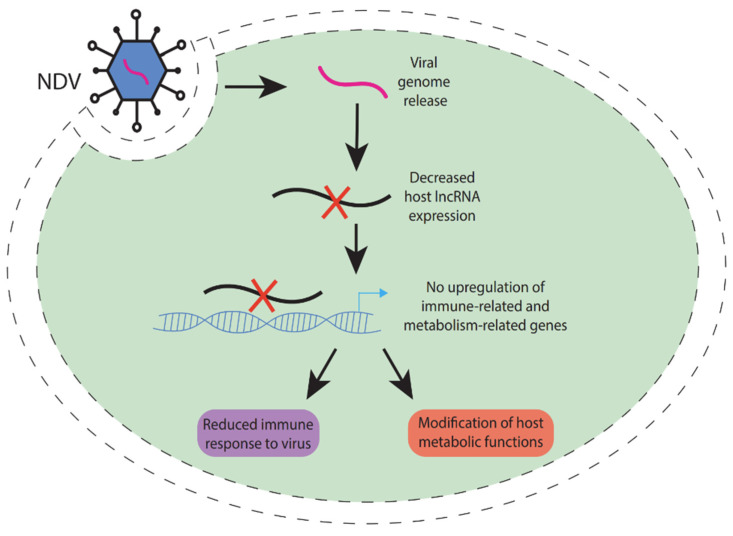
Action of lncRNAs in NDV infection [121,122]. This image illustrates the mechanism by which Newcastle disease virus (NDV) infection affects host cellular processes. Upon viral genome release into the host cell, there is a decrease in host lncRNA expression. This leads to no upregulation of immune-related and metabolism-related genes, resulting in a reduced immune response to the virus and modification of host metabolic functions.

**Table 1 ncrna-11-00042-t001:** Examples of lncRNA regulating IAV replication.

lncRNA	Influenza Strain	Mechanism	Subcellular Localization	Potential Application	References
NRAV	A/WSN/1933 (H1N1)	Suppresses the initial transcription of a number of important ISGs, including MxA, IFITM3, OASL, IFIT2, and IFIT3.	Nucleus	Potential antiviral target	Ouyang et al., 2014 [140]
TSPOAP1- AS1	A/Puerto Rico/8/1934 (H1N1)	OASL, ISG20, IFIT1, IFITM1, and other anti-IAV ISGs are negatively regulated, which suppresses IAV-triggered type I IFN signaling.	Nucleus	Potential antiviral target	Wang Q et al., 2022 [141]
lnc-Lsm3b	A/Puerto Rico/8/1934 (H1N1)	Blocks overproduction of type Ά IFNs and inhibits RIG-I activation by competing with viral RNAs for the binding of RIG-I monomers.	Cytoplasm	Potential antiviral target	Jiang et al., 2018 [143]
IPAN	A/WSN/1933 (H1N1)	Enhances the stability of viral PB1 by forming an association that promotes IAV transcription and replication.	Cytoplasm/Nucleus	Potential antiviral target	Wang et al., 2019 [144]
lncRNA-PAAN	A/WSN/1933 (H1N1)	Increases viral RNA polymerase activity by facilitating assembly of the RdRp complex.	Nucleus	Potential antiviral target	Wang J et al., 2018 [145]
lncRNA-ACOD1	A/Puerto Rico/8/1934 (H1N1)	Increases the synthesis of metabolites and the catalytic activity of GOT2.	Cytoplasm	Potential antiviral target	Wang P et al., 2017 [112]
VIN	A/WSN/1933 (H1N1)	Restricts IAV replication and viral protein synthesis.	Nucleus	Increase expression using small molecule agonist	Winterling et al., 2014 [146]
lnc-PINK1-2	A/Puerto Rico/8/1934 (H1N1)	Upregulates TXNIP and reduces IAV replication.	Nucleus	Increase expression using small molecule agonist	Pushparaj et al., 2022 [147]
lncRNA DFRV	A/Beijing/501/2009 (BJ501, H1N1)	Positively regulates IL-1β and TNF-α and inhibits viral replication.	Nucleus	Potential antiviral target	Wang et al., 2023 [148]

## Data Availability

No new data were created or analyzed in this study. Data sharing is not applicable to this article.

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
