# Peer review of "An Emphasis on the Role of Long Non-Coding RNAs in Viral Gene Expression, Pathogenesis, and Innate Immunity in Viral Chicken Diseases"

_ncrna, 2025, doi:10.3390/ncrna11030042_

Round 1

Reviewer 1 Report

Comments and Suggestions for Authors

This review provides a comprehensive exploration of the role of long noncoding RNAs (lncRNAs) in chicken viral diseases, focusing on their regulatory functions in host immune responses and viral replication. The manuscript covers a wide range of viral diseases affecting poultry, including Avian Leukosis, Marek’s Disease, Infectious Bursal Disease, Avian Influenza, Infectious Bronchitis, and Newcastle Disease. It also discusses the biogenesis, classification, and mechanisms of action of lncRNAs, as well as methods for their detection.

1.      The title and abstract introduce the role of long noncoding RNAs (lncRNAs) in chicken viral diseases, highlighting their regulatory functions in immune responses and viral replication. The abstract provides an overview of the review but lacks depth in explaining the novelty and significance of the findings.

ü  The title: consider adding key lncRNAs or mechanisms (e.g., "Role of Long Noncoding RNAs in Chicken Viral Diseases: Insights into Immune Regulation and Viral Pathogenesis").

ü  Abstract:

·       The abstract mentions several lncRNAs (e.g., ERL lncRNA, linc-GALMD3, loc107051710) without providing enough context on their specific roles or significance.

·       The phrase "paving the way for future research", Replace with a more concrete statement about the potential applications of lncRNA research in poultry health.

2.      Introduction

ü  The introduction discusses the global importance of poultry farming and the challenges posed by viral diseases. It briefly introduces lncRNAs as regulators of immune responses but lacks a clear transition to their role in viral infections.

ü  The introduction does not clearly connect these challenges to the role of lncRNAs. A stronger transition is needed.

ü  The section on viral diseases, focus more on how these diseases specifically impact poultry immune responses and why lncRNAs are relevant.

ü  The introduction does not explicitly state the objectives of the review. Add a clear statement about the review's goals (e.g., summarizing current knowledge, identifying research gaps, proposing future directions).

ü  "prey of viral diseases" – Replace "prey" with "victims" or "targets."

ü  "immunosuppressive additives in poultry feed" – Clarify whether these additives are intentionally immunosuppressive or have unintended effects.

3.      Long Non-Coding RNA (lncRNA section provides a detailed overview of lncRNA classification, biogenesis, and functions, including their roles in chromatin remodeling, transcription regulation, and post-transcriptional modifications. However, the section is overly technical and lacks focus on viral diseases.

ü  The section is dense and would benefit from subheadings (e.g., "Biogenesis of lncRNAs," "Mechanisms of lncRNA Action," "lncRNAs in Viral Infections") to improve readability.

ü  The discussion on lncRNA classification (e.g., bidirectional, antisense, intergenic) is too technical for a general audience. Simplify or provide more context.

ü  The section on lncRNA functions is comprehensive, condense subsections (e.g., chromatin structure regulation) to maintain focus on viral diseases.

ü  "RNA-chromatin association detection" – Rephrase to "detection of RNA-chromatin associations."

ü  "R-loops, which are considered threats to genome stability" – Replace "considered threats" with "can pose risks to genome stability."

4.      Role of lncRNAs in Specific Chicken Viral Diseases section explores the role of lncRNAs in specific viral diseases affecting chickens, including Avian Leukosis, Marek’s Disease, Infectious Bursal Disease, Avian Influenza, Infectious Bronchitis, and Newcastle Disease. The discussion is descriptive but lacks critical analysis.

ü  The section lacks critical evaluation. For example, the discussion on Marek’s disease and lncRNAs (e.g., ERL lncRNA, linc-GALMD3) is descriptive but does not critically evaluate the strength of the evidence or highlight conflicting findings.

ü  The subsections on different diseases are uneven in depth. Some diseases (e.g., Marek’s disease) are discussed in detail, while others (e.g., Newcastle Disease) are treated more superficially

ü  The role of lncRNAs in viral replication and immune evasion is underdeveloped. Expand the discussion of how lncRNAs interact with viral proteins or host immune pathways.

ü  "DF-1 CELLS" – Use "DF-1 cells" for consistency.

ü  "loc107051710" – Introduce this lncRNA with a brief description of its function early in the section.

5.      Methods of Detecting lncRNA section reviews techniques for detecting lncRNAs, including full-length cDNA sequencing, chromatin state maps, and RNA sequencing. However, it lacks critical evaluation of the methods and does not address emerging technologies.

ü  The section lacks critical evaluation of the methods. Discuss the limitations of RNA sequencing (e.g., difficulty in detecting low-abundance lncRNAs).

ü  The discussion on chromatin state maps is too technical. Simplify for a broader audience.

ü  The section does not address emerging technologies (e.g., single-cell RNA sequencing) that could enhance lncRNA detection and functional analysis.

ü  "Cap-analysis of gene expression (CAGE) tag sequences (FANTOM3)" – Remove redundant parentheses. Use "Cap-analysis of gene expression (CAGE) tag sequences from FANTOM3."

6.      The conclusion briefly summarizes the role of lncRNAs in chicken viral diseases and their potential as therapeutic targets. However, it is too brief and does not fully synthesize the key findings of the review.

ü  The conclusion is too brief and does not fully synthesize the key findings. Expand to summarize the main points and emphasize the implications for future research.

ü  The statement "LncRNAs have a lot of potential for usage as therapeutic targets" is overly optimistic. Provide a more balanced assessment of the challenges and limitations.

ü  The conclusion does not address the translational potential of lncRNA research in poultry health management. Propose specific applications (e.g., diagnostic tools, vaccines) and highlight the steps needed to achieve them.

ü  "lncNRAs" – Correct to "lncRNAs."

ü  "serum from the chickens" – Rephrase to "chicken serum."

7.      Figures:

ü  Figure 1: Protocol of RNA Sequencing Technique

·       Add more detailed labels or annotations to explain each step of the RNA-seq process.

·       Include a brief description of how RNA-seq is used to identify and quantify lncRNAs, as this is the primary focus of the manuscript.

·       Consider using color coding or arrows to guide the reader through the workflow.

ü  Figure 2: Action of lncRNA of IBDV Under In Vitro Conditions

·       Add more detail to the figure, such as molecular interactions (e.g., binding sites, signaling pathways) between loc107051710, IRF8, and antiviral genes.

·       Include annotations or a brief description to explain the key steps in the regulatory process.

·       Consider adding a panel showing the broader context of IBDV infection and how this lncRNA-mediated response fits into the overall immune response.

ü  Figure 3: Mechanism of lncRNA Acting as Positive Regulator for Viral Replication

·       Add more detail to the figure, such as molecular interactions (e.g., binding sites, signaling pathways) between lncRNAs and host/viral components.

·       Include annotations or a brief description to explain the key steps in the regulatory process.

·       Consider adding a panel showing the broader implications of these lncRNA-mediated mechanisms for viral pathogenesis and disease outcomes.

ü  Figure 4: Action of lncRNA in NDV Infection

·       Add more detail to the figure, such as molecular interactions (e.g., binding sites, signaling pathways) between lncRNAs and host metabolic/immune components.

·       Include annotations or a brief description to explain the key steps in the regulatory process.

·       Consider adding a panel showing the broader implications of these lncRNA-mediated mechanisms for NDV pathogenesis and disease outcomes.

Comments on the Quality of English Language

The English could be improved to more clearly express the research

Author Response

Dear Reviewer,
Thank you for taking the time to carefully review the manuscript. We have addressed all the comments given. Please see the attached document for the responses. Thank you.

Reviewer 2 Report

Comments and Suggestions for Authors

The authors present a comprehensive review concerning the role of lncRNAs in avian viral diseases and discuss the prospects for enhanced diagnostic and therapeutic tools. Due to its significance for livestock economics, this review is of interest to the readership of ncRNA.

Some amendments are recommended before acceptance:

1. The authors may consider the role of extracellular vesicles in viral infections and concisely discuss the connection of lncRNAs and EVs as well  (e.g. https://doi.org/10.3390/pathogens9110876), keeping in mind the similarity of EVs and viral particles.

2. There are only a few references from 2022-2025. Any recent refs. should be considered, if suitable.

3. As a minor point: Though definitely not plagiarism, the wording of the first paragraph in Ginn et al. (2020) Diverse roles of long non-coding RNAs in viral diseases. https://doi.org/10.1002/rmv.2198, is similar to the first paragraph of the Conclusions. The wording may be slightly altered.

Author Response

Dear Reviewer,
Thank you for taking the time to carefully review the manuscript. We have addressed all the comments given. 

Comment 1: The authors may consider the role of extracellular vesicles in viral infections and concisely discuss the connection of lncRNAs and EVs as well (e.g. https://doi.org/10.3390/pathogens9110876), keeping in mind the similarity of EVs and viral particles. 

Response 1: 

  • Added new information in the introduction section. Line no. 100 -135. Page no. 3-4

Comment 2: There are only a few references from 2022-2025. Any recent refs. should be considered, if suitable. 
Response 2: Added new relevant references

Comment 3: As a minor point: Though definitely not plagiarism, the wording of the first paragraph in Ginn et al. (2020) Diverse roles of long non-coding RNAs in viral diseases. https://doi.org/10.1002/rmv.2198, is similar to the first paragraph of the Conclusions. The wording may be slightly altered. 
Response 3: Updates were done to rectify the comments.

Reviewer 3 Report

Comments and Suggestions for Authors

In the manuscript titled “Role of long noncoding RNAs in chicken viral diseases”, the authors explores the roles of long noncoding RNAs (lncRNAs) in various viral diseases affecting poultry, such as avian leukosis, Marek’s disease, infectious bursal disease, avian influenza, infectious bronchitis, and Newcastle disease. They highlighted how lncRNAs influence gene expression at multiple levels, including chromatin remodeling, transcription, and post-transcriptional regulation. The review discusses specific lncRNAs associated with immune responses and viral pathogenesis, and their potential as diagnostic and therapeutic tools for poultry health management. The manuscript also underscores the need for further investigation into mechanisms by which lncRNA function in host-virus interactions and poultry disease resistance. Following are my specific concerns.

Specific comments

1.    The manuscript provides an overview of lncRNAs' roles in viral diseases, but lacks in-depth mechanistic details for some of the described lncRNAs. For instance, the pathways through which ERL lncRNA modulates immune responses during Marek's disease need further elaboration. Including the details would enhance the scientific depth of the review.

2.    The conclusion briefly mentions the potential of lncRNAs as diagnostic tools and therapeutic targets. Recently, research on non-coding RNAs has expanded, with increasing attention given to both lncRNAs and circRNAs. Including in the Conclusions section that circRNAs have therapeutic potential and that investigating their relationship with this review’s points will be an important avenue for future research. Add the point and cite the following paper: Ju et al. PMID: 36697235.

3.    The tables summarizing lncRNA functions could include additional columns for "Proposed Mechanisms" and "Potential Applications," providing a more structured overview.

Author Response

Dear Reviewer,
Thank you for taking the time to review the manuscript. All the comments were addressed. Thank you.

Comment 1: The manuscript provides an overview of lncRNAs' roles in viral diseases, but lacks in-depth mechanistic details for some of the described lncRNAs. For instance, the pathways through which ERL lncRNA modulates immune responses during Marek's disease need further elaboration. Including the details would enhance the scientific depth of the review.

Response 1: Relevant information was added. (line no. 724-732, page no. 21; Line no. 751-771, page no. 22)

Comment 2: The conclusion briefly mentions the potential of lncRNAs as diagnostic tools and therapeutic targets. Recently, research on non-coding RNAs has expanded, with increasing attention given to both lncRNAs and circRNAs. Including in the Conclusions section that circRNAs have therapeutic potential and that investigating their relationship with this review’s points will be an important avenue for future research. Add the point and cite the following paper: Ju et al. PMID: 36697235. 
Response 2: Thank you for the valid comments. We have updated the conclusion and added the recommended citation. Thank you.

Comment 3: The tables summarizing lncRNA functions could include additional columns for "Proposed Mechanisms" and "Potential Applications," providing a more structured overview.
Response 3: Thank you for your comment. We have added the Potential applications column in the table.

Round 2

Reviewer 3 Report

Comments and Suggestions for Authors

The authors addressed my previous concerns successfully.